# Assessment of Cardiometabolic Risk Factors, Physical Activity Levels, and Quality of Life in Stratified Groups up to 10 Years after Bariatric Surgery

**DOI:** 10.3390/ijerph16111975

**Published:** 2019-06-04

**Authors:** Larissa Monteiro Costa Pereira, Felipe J. Aidar, Dihogo Gama de Matos, Jader Pereira de Farias Neto, Raphael Fabrício de Souza, Antônio Carlos Sobral Sousa, Rebeca Rocha de Almeida, Marco Antonio Prado Nunes, Albená Nunes-Silva, Walderi Monteiro da Silva Júnior

**Affiliations:** 1Post Graduate Program in Physical Education, Federal University of Sergipe, São Cristovão, SE 49100-000, Brazil; larissa_monteiroo@hotmail.com (L.M.C.P.); fjaidar@gmail.com (F.J.A.); walderim@yahoo.com.br (W.M.S.J.); 2Department of Physical Education, Federal University of Sergipe, São Cristovão, SE 49100-000, Brazil; raphaelctba20@hotmail.com; 3Group of Studies and Research of Performance, Sport, Health and Paralympic Sports—GEPEPS, Federal University of Sergipe, São Cristovão, SE 49100-000, Brazil; 4Post Graduate Program in Health Sciences, Federal University of Sergipe—UFS, São Cristovão, SE 49100-000, Brazil; jadernetofisio@hotmail.com (J.P.d.F.N.); rebeca_nut@hotmail.com (R.R.d.A.); nunes.ma@outlook.com (M.A.P.N.); 5Post Graduate Program in Physiological Sciences, Federal University of Sergipe, São Cristovão, SE 49100-000, Brazil; 6Department of Physical Therapy, UniversityHospital, Federal University of Sergipe, Aracaju, SE 49100-000, Brazil; 7Department of Medicine, Federal University of Sergipe, São Cristovão, SE 49100-000, Brazil; acssousa@terra.com.br; 8Exercise’s Inflammation and Immunology Laboratory, Sports Center, Federal University of Ouro Preto, Ouro Preto, MG 35400-000, Brazil; albenanunes@hotmail.com

**Keywords:** bariatric surgery, physical activity, health, quality of life

## Abstract

Obesity is a highly prevalent chronic metabolic disease, with an increasing incidence, and is currently approaching epidemic proportions in developing countries. Ouraim was to evaluate the activity levels, quality of life (QoL), clinical parameters, laboratory parameters, and cardiometabolic risk factors afterbariatric surgery (BS). We classified78 patients who underwentBS into four groups, as follows: Those evaluated 1–2 years after BS (BS2), 2–4 years after BS (BS4), 4–6 years after BS (BS6), and 6–10 years after BS (BS+6). Body weight (BW), body mass index (BMI), comorbidities associated with obesity (ACRO), physical activity level, and QoL were evaluated. Patients exhibited improvements in BW, BMI, cardiometabolic risk, hypertension, dyslipidemia, and diabetes and significant changes in lipid profiles in the first postoperative yearafter BS.The physical activity level inthe BS2, BS4, and BS6 groups was increased, compared with that in the first postoperative year, with a decrease in International Physical Activity Questionnaire scores at 1 year in the BS2 (207.50 ± 30.79), BS4 (210.67 ± 33.69), and BS6 (220.00 ± 42.78) groups. The QoL of patients in theBS2 and BS4 groups was excellent and that of patients in the BS4 and BS+6 groupswas very good. These findings suggest that BS promoted improved physical activity levels and QoL and reduced comorbidities in patients with morbid obesity.

## 1. Introduction

Obesity is a highly prevalent chronic metabolic disease with an increasing incidence and is currently approaching epidemic proportions in developing countries [1]. Excess body weight may be associated with the development of systemic arterial hypertension (SAH), type 2 diabetes mellitus (DM), and dyslipidemia, all of which are considered to be cardiometabolic risk factors [2].

In Brazil, obesity affects 18.9% of the population, i.e., almost one in five Brazilians [3]. Following the Brazil Ministry of Health order [3], as of 19 March 2013, surgical interventions for morbid obesity were formalized and performed by the Unified Health System, with Brazil being the second country in the world to routinely perform bariatric surgery (BS) [4]. BS appears to be an effective alternative for the treatment of obesity and reduces the risks associated with comorbidities [5].

Conservative treatment, diet, and exercise have been proven to induce significant weight loss and increase aerobic fitness with cardiometabolic benefits [6]. However, the low long-term adherence of individuals to these approaches may lead to the recovery of body weight [7]. The bariatric surgery procedure does not minimize the risks associated with a sedentary lifestyle [8] and a lifestyle change after surgery is essential for optimal outcomes [9,10].

The incidence of sedentary lifestyles before BS is high and, despite the weight loss achieved postoperatively, most individuals continue to be inactive [11,12]. The practice of engaging in regular physical exercise is one of the predictors of weight loss after BS [13,14]. Physically active individuals have a lower body mass index (BMI) and achieve greater weight loss [15,16].

Physical activity improves glucose metabolism, body composition, and quality of life (QoL) [17,18]. Psychosocial interventions, such as cognitive-behavioral therapy, are effective not only in improving adherence but also in disseminating information about the disease and how to deal with it [19].

Individuals with obesity generally have a greater difficulty in performing daily physical activities, which is associated with a lower QoL [20,21]. Some studies reported an improvement in QoL after BS, as measured by generic QoL instruments, such as the Short-Form 36, the Gastrointestinal Life Quality Questionnaire, and the Bariatric Analysis and Reporting Outcome System (BAROS)questionnaire [22,23]. However, other studies suggest that patients who recovered from BS are at greater risk of exhibiting emotional behavior and depressive symptoms [21,24].

Psychosocial variations can affect the involvement of individuals in engaging in physical activity, such as underestimating their physical capacity based on exerting low efforts [25,26]. In general, the prevalence of a sedentary lifestyle is higher in obese individuals and may be associated with lower social interest [27,28]. Psychological and behavioral factors associated with BS continue to receive little attention [29].

Therefore, the primary objective of the present study was to evaluate the changes in the individuals’ activity level, QoL, clinical parameters, laboratory parameters, and cardiometabolic risk factors in groups stratified by the length of time since BS, from 1 to 10 years.

## 2. Materials and Methods

The research followed the components of the Protocol Strengthening the Reporting of Observational Studies in Epidemiology (STROBE). Figure 1 illustrates the procedures followed in this study.

This was a cross-sectional study with retrospective data extracted from medical records. Biochemical data were acquired from medical requests in the postoperative follow-up of the multi-professional service of BS of the University Hospital of the Federal University of Sergipe.

### 2.1. Sample

The study included 78 patients, undergoing BS, attending the University Hospital of Sergipe, Sergipe, Brazil over a 10 year period, accompanied by a multi-professional team from the bariatric ambulatory service. The sample represents subjects undergoing BS through the public health system in the only public hospital that performs such procedures in the state of Sergipe, Brazil. Notably, Brazil’s health system is divided into the public sector, which meets 69.70% of demands, and the private sector, which serves the remaining population [3].

The patients in this study were divided into four groups, as follows: Patients assessed between 1 and 2 years after BS (BS2), between 2 and 4 years after BS (BS4), between 4 and 6 years after BS (BS6), and between 6 and 10 years after BS (BS+6) (Table 1).

The inclusion criteria were as follows: The subject underwent BS at least 1 year prior, was aged between 18 and 60 years, and remained in follow-up postoperative care at the University Hospital of Sergipe. Those who did not participate in any phase of follow-up were excluded from the study. This was a mixed retrospective and prospective study and the International Physical Activity Questionnaire (IPAQ) and BAROS questionnaires, as well as anthropometric data, were used. All patients under went the same surgical procedure by gastric bypass, which was an inclusion criterion. Demographic evaluation of the cohort indicates a reduction in the number of individuals undergoing BS over the years, reflecting the aggravation of the economic crisis in Brazil in 2014, with immediate repercussions on financial transfers to health by the federal government. All procedures were performed at a federal university hospital, which is the only hospital in the state of Sergipe equipped to perform the surgical procedure for the treatment of obesity.

Biochemical data from the preoperative period up to 1 year after BS, following the intervals described in the table and accompanying anthropometric measures during the time, were acquired from medical records, performed at the present time, IPAQ and BAROS questionnaires, and anthropometric measurements.

All patients provided written informed consent in accordance with Resolution 510/2016 of the National Health Council, a research regulatory standard involving the use of data. This study was approved by the Research Ethics Committee of the University Hospital of Aracaju, Federal of Sergipe, on 7 August 2017 under the protocol number 2,203,872.

The city of Sergipe is one of the 27 federative units of the Federative Republic of Brazil. It is situated in the Northeast Region and borders the Atlantic Ocean to the east; the state of Bahia to the west and south; and the state of Alagoas to the north, from which it is separated by the São Francisco River. It is the smallest Brazilian state and occupies a total area of approximately 21,918 km^2^, which is slightly larger than Israel. It has an estimated population of 2,278,308 inhabitants, in 2018, with a Human Development Index of 0.665.

### 2.2. Data Collection (Anthropometric Data)

Weight (admission, preoperative, postoperative, minimum, and current), height, and BMI (calculated as weight (kg) by height squared (m^2^)) were classified according to cutoff points proposed by the World Health Organization [30] for nutrition service records. Excess body mass was calculated as the difference between pre-surgical weight and ideal weight according to the Metropolitan Life Foundation Table. For the calculation of the percentage of excess weight loss (%EWL), the following equation was used: %EWL = ((initial weight − actual weight/usual weight − ideal weight) × 100).

### 2.3. Biochemical Data

Biochemical data, i.e., serum and/or plasma triglycerides, total cholesterol, HDL-cholesterol, LDL-cholesterol, and fasting glycemia levels, were obtainedfrom the University Hospital of Sergipe medical records at four time points, as follows: Admission, release (3 months), postoperative (PO) (6 months), and final (12 months).

### 2.4. Evaluation of Comorbidities Associated with Obesity (ACRO)

Silva-Neto et al. [31] proposed an instrument capable of quantitatively measuring the changes in cardiometabolic risk of obesity-related comorbidities after BS. The instrument enables quantification of the improvement or reversal of cardiometabolic risk reduction (CRR) components. The scores for the Evaluation of Comorbidities Related to Obesity (ACRO) were calculated based on a points system that assigned scores of 0–5, according to severity, for the following CRR components: DM, dyslipidemia, and SAH (Table 2). The scores were assigned at the time of admission, release after surgery, and postoperative follow-up (approximately 3, 6, and 12 months) [32].

In this study, a cut off score of ≤2 indicated the absence of comorbidities, whereas a score of ≥3 indicated the presence of comorbidities (Diabetes, dyslipidemia, and hypertension) [32].

### 2.5. Data on Quality of Life

The QoL questionnaire was administered at the first interview for patients in the postoperative period after BS, according to Oria and Moore head [33], and we ascertained their current weight for later calculation of the percentage of loss that is part of the BAROS instrument. Weight loss was ascertained by measuring the differences in weight and BMI between the pre- and postoperative periods and the percentage of excess weight loss (%EWL) in the postoperative period. BMIs were classified according to the adapted World Health Organization (WHO) tables. Completion of these tables and the questionnaire measured improvements and/or control of comorbidities and QoL (self-esteem, physical, social, professional and sexual activity, complications, and reoperations). Based on their final scores, patients were classified into one of the following groups, which received the appropriate designation for postoperative progress: Insufficient, acceptable, good, very good, or excellent.

### 2.6. International Physical Activity Questionnaire (IPAQ)

The short format of the IPAQ was used at three time points (admission, after 1 year, and the current time) to examine the level of physical activity. The questionnaire was structured and comprised questions regarding the frequency and duration of physical activities. The individuals were classified as very active, active, irregularly active, and sedentary [34].

### 2.7. IPAQ Classification

The questionnaire contained questions related to physical activities performed in the last week prior to the application of the questionnaire. The responses of the individuals were analyzed and the individuals were classified according to the questionnaire, which divides and conceptualizes the categories as follows:

*Sedentary*: Individuals who do not perform any physical activity for at least 10 min continuously during the week;

*Insufficiently Active*: Individuals who engage in physical activity for at least 10 min continuously per week, but insufficiently to be classified as active. For inclusion in this category, the duration and frequency of the different types of activities (moderate + vigorous + walks) are added. This category is divided into the two following groups:

*Insufficiently Active A*: Individuals who perform 10 min of continuous physical activity, adhering to at least one of the following criteria: A frequency of 5 days/week or a duration of 150 min/week;

*Insufficiently Active B*: Individuals who do not meet any of the specifications of the individuals in the Insufficiently Active A category.

*Active*: Individuals who meet the following recommendations: (a) Vigorous physical activity ≥3 days/week and >20 min/session; (b) moderate exercise or walking ≥5 days/week and >30 min/session; (c) any activity added to> 5 days/week and >150 min/week;

*Very Active*: Individuals who meet the following recommendations: (a) Vigorous ≥5 days/week and >30 min/session; (b) vigorous ≥3 days/week and >20 min/moderate + and/or a walk 3 to 5 days/week for a >30 min/session.

### 2.8. Study

Retrospective data was collected via a survey of medical records of the University Hospital of Sergipe and the questionnaires were administered by a single trained evaluator. QoL and physical activity levels were assessed through questionnaires and considered the level of physical activity, which was classified as physically active (active) or insufficiently active (IA). For the purpose of this study, the active group comprised individuals who classified themselves as very active or active in the IPAQ questionnaire, and the respondents who reported themselves as IA, IA-A, IA-B, and sedentary were classified as belonging to the IA category.

### 2.9. Statistics

Statistical analysis was performed, to assess the relationship between the groups, using the Statistical Package for Social Science, version 22.0 (SPSS Inc., Chicago, IL, USA). The central tendency measures, mean ± standard deviation (X ± SD), were used. To verify the normality of the variables, the Shapiro–Wilk test was used, considering the sample size. One-way analysis of variance (ANOVA) and the Bonferroni post-hoc test for BAROS and EWL in the groups was used to verify the possible differences between groups divided by BS postoperative time. For the other indicators analyzed, two-way ANOVA (Group × Moments) and Bonferroni post-hoc tests were used. In relation to ACRO, individuals were counted in relation to the cutoff point. To verify the effect size, the Cohen *f*^2^ test was used in addition to the cutoff points, selected at 0.02 to 0.15 as a small effect, from 0.15 to 0.35 as a medium effect, and greater than 0 as a large effect [35]. A *p*-value of *p* < 0.05 was considered statistically significant.

## 3. Results

The study population comprised more women than men in all groups, BS2 (83.3%), BS4 (64.3%), BS6 (86.4%) and BS+6 (73.3%). The weight loss achieved at various time points, i.e., weight at admission, weight after 3 months of multi-professional care and before BS, postoperative weight immediately after BS, at postoperative 6 months, and weight at final evaluation (12 months), is shown in Table 1. The current body weight was significantly reduced in BS2, BS4, and BS6. The BS+6 group individuals had the highest weight in both the pre- (134.36 ± 29.60 kg) and postoperative (124.50 ± 28.42 kg) measurements.

Following BS, significant reductions in the waist circumference of all groups, were observed, indicating a possible reduction of cardiovascular risk associated with visceral adiposity.

The BMI in all groups at admission was higher than 40 kg/m^2^ (Figure 4), and a significant reduction in the current BMI was observed in groupsBS2, BS4, and BS6, with BS+6 being the group with the highest BMI both pre- and postoperatively.

The level of physical activity increased after the first postoperative year and was significant for BS2, BS4, and BS6, with a sedentary classification for exercise and a subsequent fall in the current IPAQ, even though the exercise classification was maintained. Additionally, the QoL was significantly different in the BS4 and BS6 groups, compared with that in the OTHER groups.

Differences in weight, waist circumference, BMI, physical activity level (IPAQ), EWL, and QoL (BAROS) are summarized in Table 3.

Changes in weight, waist circumference, BMI, physical activity level (IPAQ), EWL, and QoL (BAROS) in different groups are shown in Figure 2, Figure 3, Figure 4 and Figure 5.

Changes in biochemical marker levels are shown in Table 4. LDL and total cholesterol levels significantly improved postoperatively in BS2, BS4, and BS+6. Triglyceride levels improved significantly in all groups and glycemia levels significantly improved in BS +6.

Table 4 shows a decline in biochemical marker levels. Over the first postoperative year, LDL and total cholesterol levels significantly declined inBS2, BS4, and BS+6, and triglycerides levels significantly declined in all groups. However, glycemia significantly declined in only BS+6.These results may be associated with immediate responses to the surgical procedure, as well as the greater practice of physical activity in the first 12 months after surgery.

The effect of surgery on cardiometabolic risk and associated pathologies were evaluated. Based on the ACRO risk score, the frequency of a score ≥3, indicating the need for chronic medical treatment or complications related to comorbidity, was higher for patients with SAH and showed a higher frequency between groups, followed by dyslipidemia and DM. These results may be associated with increased weight, waist circumference (WC) and, consequently, increased BMI.

The evolution of factors associated with cardiometabolic risk at various time points in the first postoperative year is described in Table 5. With regard to dyslipidemia and diabetes, there was a significant reduction in the frequency of patients using at least one drug after BS.

## 4. Discussion

A predominance of women was observed in our study, which is consistent with previous findings [36,37]. Women of all ethnic groups have a greater acceptance of undergoing BS compared with men [38]. Social, economic, and cultural motivations explain this prevalence, which is associated with a lower acceptance of healthcare by men [39]. Women accounted for 75–80% of individuals in previous cohorts in studies of BS [40,41].

BMI and EWL are important clinical variables for evaluating long-term weight maintenance [42]. In the present study, the groups showed reductions in BMI and satisfactory EWL, which is consistent with the results of systematic reviews and goal-analyses [43]. The first year after surgery tends to represent the greatest weight loss of an individual [44], this pattern was observed in group BS2, BS4, BS6, and BS+6 individuals, all of whom achieved minimum weight loss during the first year postoperatively.

Surgical success is accepted as 50% EWL [45], a result found in all groups, but considering the changes in BMI classification, only BS6 showed satisfactory results, i.e., BMI < 30 kg/m^2^, with a modification of the BMI classification of morbid obesity [46].

The highest cardiovascular risk is associated with visceral adiposity, making WC the most sensitive measure [47,48]. In BS2, BS4, BS6, and BS+6, this risk was substantially reduced in all groups using a cutoff value of 88 cm for women and 102 cm for men [30]. Andersson et al. [49] stated that WC can be used to evaluate the reversion of insulin resistance in obese patients after weight loss following BS.

The increased interest in exercise during the first postoperative year is associated with the weight loss achieved [50,51,52,53,54] and our findings showed that exercise was higher during this period, in all groups, and was significantly different for groups BS2, BS4, and BS6. Current guidelines recommend 150 min of moderate activity per week, or 10,000 steps/day for adults [55], and over 200 min per week of moderate intensity activity to prevent weight gain and ensure significant weight loss [56].

Improvements in post operative physical activity levels have been previously reported [57]. In the United States, United Kingdom, and Sweden, it has been noted that most individuals improved their sedentary behavior following BS, despite an increase in exercise [57,58,59]. Our data demonstrated that the current level of physical activity in the groups was lower, compared with that in the first postoperative year, but the classification of activity level was maintained in all groups.

Bobowicz et al. [60] used BAROS to evaluate 84 patients 5 years after surgery and the best results were associated with greater weight loss and physical, professional, social, and sexual improvements. Similar to the above findings, in our study, groups with a higher percentage of EWL had higher levels of physical activity and were more likely to classify their current QoL after surgery in the BS2 and BS6 groups as excellent (4.73 and 4.93, respectively) and very good in BS4 (3.10) and BS+6 (4.11) groups.

The Roux-en-Y gastric bypass (RYGB) is associated with improvements in biochemical indicators, such as triglycerides, total cholesterol, and LDL-cholesterol, which have been attributed to weight loss [61,62,63]. In the present study, the groups that achieved greater weight loss had better serum lipid profiles.

Rêgo et al. [64] assessed 134 patients after undergoing BS and noted changes in metabolic parameters and significant reductions in LDL-cholesterol, total cholesterol, and triglyceride levels in men and women. Our data corroborate the findings of these studies and individuals in all groups had better lipid profiles 12 months after BS.

Sustained weight loss, improved lipid profile, and lower risk of cardiovascular disease were noted in all follow-up groups (BS2, BS4, BS6, and BS+6). Similar results were found in a study of 1048 obese individuals [65], wherein the authors concluded that the improvement in the lipid profile is proportional to amount of weight loss achieved, being higher in the first 12 months, and is maintained for up to 5 years.

Our study demonstrated that the best glycemic profiles were found in the BS+6 group, with a longer postoperative time. Others studies that evaluated BS and biochemical indicators of short-term glycemia [66,67,68] showed significant remission rates for type 2 diabetes and unsatisfactory improvements in biochemical profiles. However, long-term studies with longitudinal follow-ups demonstrated reductions in the remission of type 2 diabetes and lower glycemic rates [69,70].

DM and SAH are the main comorbidities associated with obesity and, in our study, a reduction in DM and SAH was correlated with EWL after BS, as indicated by the ACRO score. Garcez et al. [71] and Carswell et al. [61] obtained results similar to ours, indicating a relationship between gradual weight reduction and improvements in metabolism, with a decrease in hypertension and DM. The reduction in the use of drugs to treat and control comorbidities associated with obesity postoperatively has been evaluated by several studies [72,73].

The control of comorbidities was performed quantitatively [38]. The ACRO instrument was applied and clinical progress was marked by a significant decrease in the average score of the comorbidities associated with obesity (DM, arterial hypertension, and dyslipidemia). The results obtained in our study are similar to those of Farin holt et al. [38], who assessed 1368 individuals with a postoperative reduction of comorbidities.

The metabolic impact of BS on the reduction of cardiovascular risk factors and the prevention of these factors has been documented in several studies [38,74,75,76]. A meta-analysis of the long-term effect of BS, DM, and hypertension showed that the risk decreased after BS, with relative risks of 0.33 (95% CI = 0.26–0.41; I^2^ = 42%), 0.54 (95% CI = 0.46–0.64, I^2^ = 68%), and 0.33 (95% CI = 0.22–0.46, I^2^ = 74%). However, all risks for cardiovascular outcomes reached a plateau 20–40 months after surgery [77].

This study has some limitations, including sample selection bias, in which data were collected retrospectively from records of patients who underwent BS, and a lack of additional data, such as blood pressure values at specific times and additional biochemical tests, such as glycosylated hemoglobin and insulin dosage. Another limitation was the lack of equity between the groups, which is a result of the non continuity of public investments in Brazilian health, a fact partly explained by the present economic crisis in Brazil.

Differences between genders were not evaluated, considering that the number of women was greater than the number of men. Postmenopausal women represented less than 30% of the total evaluated population and some groups did not include postmenopausal women. Therefore, such an evaluation was unfeasible. Estrogen deficiency tends to promote alteration of body composition, particularly increasing abdominal obesity during menopause [78]. Additionally, the increasing prevalence of obesity and related comorbidities require early identification and management. Thus, changes in lifestyle, such as balanced diet and physical activity, are essential to improve metabolic abnormalities and achieve a decrease in subsequent cardiovascular risk [79].

We did not evaluate the possible reduction of growth hormone (GH). However, the low level of GH in obese individuals might be associated with an increase in the prevalence of risk factors and detrimental changes in body composition that contribute to worsening their cardiometabolic risk profile [80]. In this sense, GH secretion is expected to increase significantly after RYGB gastric bypass [81], in which the reduction of cardiometabolic risk might be associated with sustained weight loss, albeit reversibly [80].

## 5. Conclusions

When evaluating the profile of patients undergoing BS in the present healthcare system, we found that the level of physical activity increased postoperatively and that QoL was excellent in the groups with a greater percentage of excess weight. Additionally, weight loss was progressively achieved with improvements in the lipid profile in the immediate postoperative period and improvements in comorbidities associated with obesity were determined via quantitative evaluation by ACRO, with a decrease in diabetes, dyslipidemia, and hypertension noted in all groups in the first postoperative year.

Furthermore, differences were found, even with a multidisciplinary approach and with objective criteria by the regulatory bodies. These differences indicate that there is a change in the control mechanisms, the control of the public health agencies, and in relation to the follow-up of the subjects undergoing BS, with a view to a prognosis more focused on what is expected.

## Figures and Tables

**Figure 1 ijerph-16-01975-f001:**
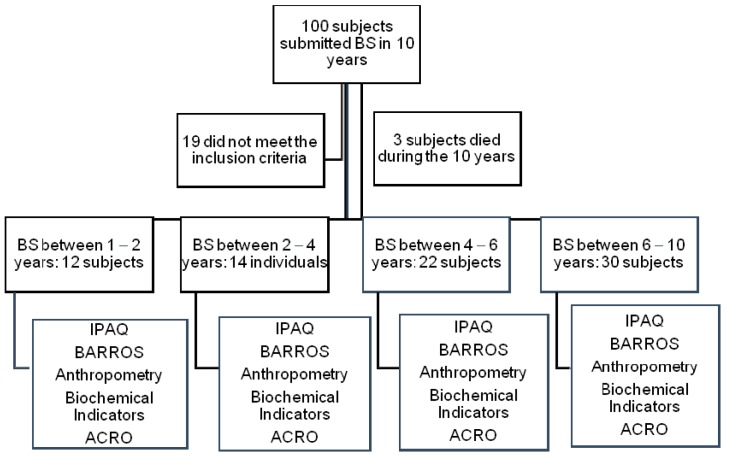
Study design. BS: Bariatric Surgery, IPAQ: International Physical Activity Questionnaire, BAROS: Bariatric Analysis and Reporting Outcome System Questionnaire, Anthropometry: weight (kg), height (cm), waist circumference (cm) (HDL, LDL), triglycerides, fasting glycemia (at four time points: admission, release, postoperative, and final), ACRO: Evaluation of Comorbidities Associated with Obesity.

**Figure 2 ijerph-16-01975-f002:**
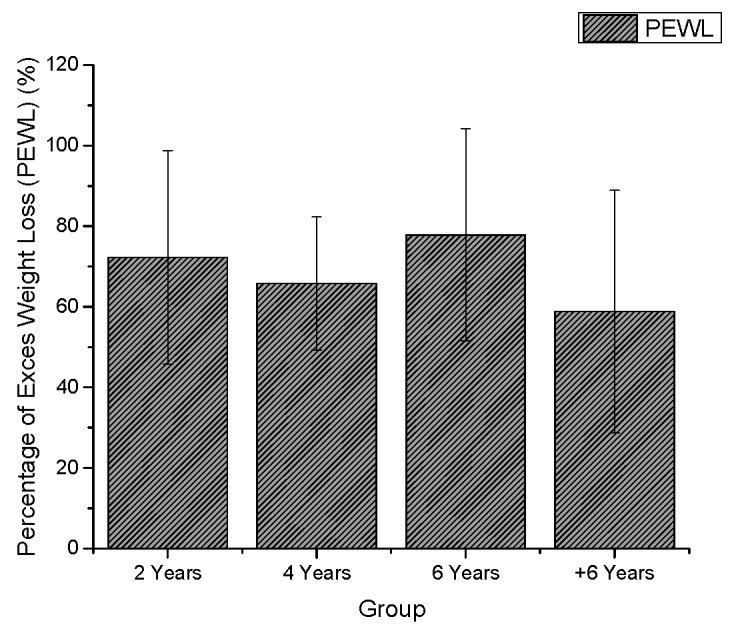
Percentage of excess weight loss (PEWL) in the different groups.

**Figure 3 ijerph-16-01975-f003:**
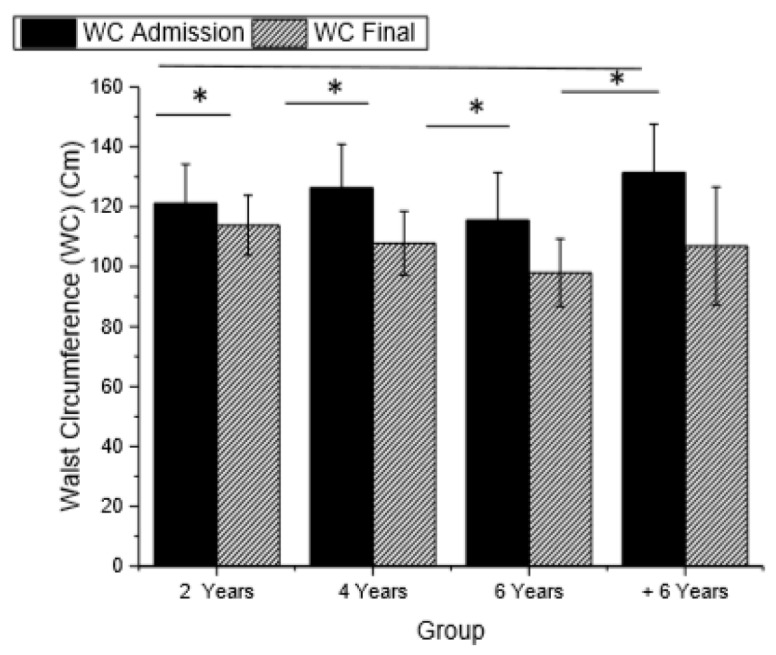
Waist circumference (WC) in the different groups (*p* < 0.001, f2 of Cohen = 0.356).

**Figure 4 ijerph-16-01975-f004:**
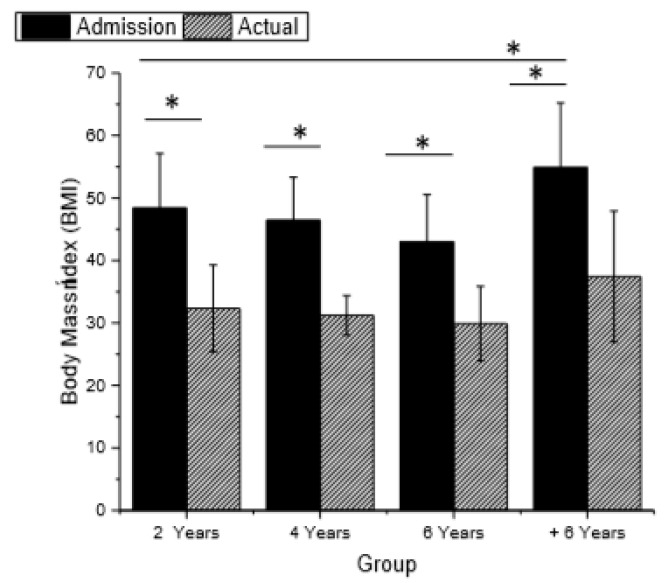
Body mass index in the different groups (*p* < 0.001, f2 of Cohen = 0438).

**Figure 5 ijerph-16-01975-f005:**
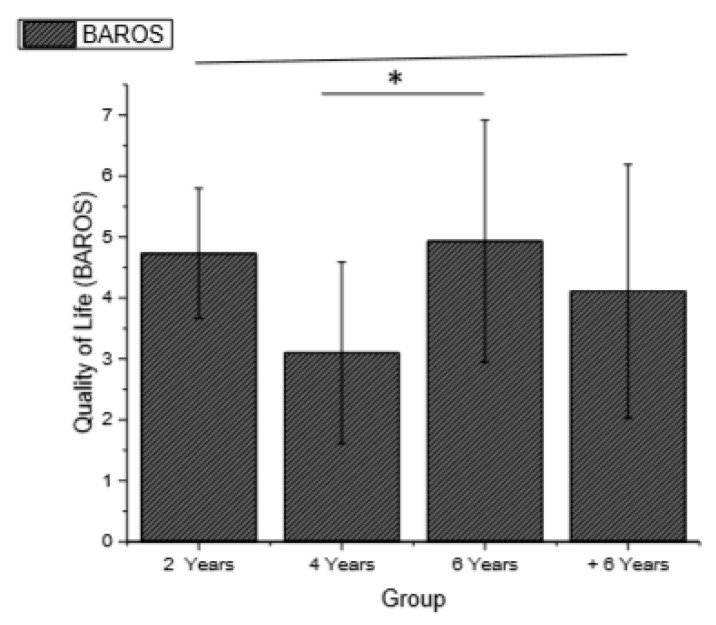
Quality of life in different groups (*p* = 0.049, f^2^ of Cohen = 0.099).

**Table 1 ijerph-16-01975-t001:** Distribution of patients in the groups (mean ± standard deviation) (Male: ♂, Female: ♀).

	BS 2 Years	BS 4 Years	BS 6 Years	BS+6 Years
Age	45.0 ± 11.0	37.0 ± 6.6	40.0 ± 11.7	40.0 ± 9.0
Age—♂	46.0 ± 4.0	36.0 ± 6.9	37.0 ± 12.8	39.0 ± 9.4
Age—♀	45.0 ± 11.5	36.0 ± 5.6	40.0 ± 11.7	40.0 ± 9.0
Gender ♂(%)/♀(%)	2(16.7)/10(83.3)	5(35.7)/9(64.3)	3(13.6)/19(86.4)	8(26.7)/22(73.3)

Note: Age refers to the time between surgery and the present time.

**Table 2 ijerph-16-01975-t002:** Evaluation of obesity-related comorbidities (ACRO).

ACRO Score	Description
**Diabetes Mellitus**	
0	Absence
1	Glucose intolerance
2	Diabetes mellitus (diagnosed)
3	Controlled with oral antidiabetic
4	Insulin therapy
5	Clinical complications
**Dyslipidemia**	
0	Absence
1	Borderline (200–239 mg/dL)
2	Conventional control (diet + physical activity)
3	Single medicinal product
4	Multiple medication
5	Uncontrolled
**Systemic arterial hypertension**	
0	Absence
1	Borderline (systolic: 130–139 mmHg, diastolic: 85–89 mmHg)
2	Conventional control (diet + physical activity)
3	Single medicinal product
4	Multiple medication
5	Uncontrolled

**Table 3 ijerph-16-01975-t003:** Differences in weight, waist circumference, body mass index, percentage of excess weight loss, physical activity level, and quality of life between the groups (mean ± standard deviation).

	BS2 *n*: 12	BS4 *n*: 14	BS6 *n*: 22	BS+6 *n*:30	*p*	*f^2^ de Cohen*
Admission weight	124.8 ± 24.6 ^a^	127.1 ± 21.0 ^a^	111.6 ± 21.8 ^a^	143.8 ± 29.3 ^a^		
Preoperative weight	120.5 ± 21.7 ^a^	120.7 ± 15.6 ^a^	108.8 ± 22.4	134.3 ± 29.6 ^a^		
Postoperative weight	111.8 ± 19.1	116.4 ± 14.0	101.6 ± 23.0	124.5 ± 28.4 ^a^		
Current weight	83.5 ± 21.1 ^ab^	85.4 ± 10.1 ^ab^	77.3 ± 16.5 ^ab^	97.2 ± 25.1		
Minimum weight	83.5 ± 21.2 ^ab^	83.8 ± 12.0 ^ab^	75.8 ± 17.2 ^ab^	94.1 ± 24.3 ^ab^	0.001	0.458 ^#^
IPAQ Admission	135.00 ± 27.47	134.67 ± 24.75	151.67 ± 25.66	109.67 ± 27.98		
IPAQ after 1 year	207.50 ± 30.79 ^a^	210.67 ± 33.69 ^ab^	220.00 ± 42.78 ^ab^	197.67 ± 58.70 ^a^		
Current IPAQ	189.17 ± 28.43	162.67 ± 29.63	193.33 ± 45.64	172.67 ± 53.24	0.001	0.438 ^#^

*p* < 0.05 (one-way ANOVA and post-hoc Bonferroni test). ^#^ Large effect. a: Different compared with other time points within the groups (*p* < 0.001). b: Different compared with other groups and time points (*p* < 0.001).BS2: Up to 2 years after BS; BS4: Up to 4 years after BS; BS6: Up to 6 years after BS; BS+6: More than 6 years after BS.

**Table 4 ijerph-16-01975-t004:** Biochemical marker levels in the groups.

	BS2	BS4	BS6	BS+6	*p*	*f^2^ de Cohen*
HDL Admission	42.33 ± 7.74	43.80 ± 13.49	42.38 ± 14.48	41.40 ± 9.82		
HDL Release	45.17 ± 9.61	42.09 ± 10.40	48.31 ± 14.57	43.13 ± 8.87		
HDL post BS	40.67 ± 9.23	42.07 ± 12.33	42.05 ± 12.68	47.30 ± 12.86		
HDL Final	47.92 ± 14.96	47.87 ± 9.45	49.19 ± 12.03	44.17 ± 12.13	0.098	---
LDL Admission	120.32 ± 42.81	117.32 ± 39.52	14135 ± 41.23 ^a^	120.49 ± 25.02		
LDL Release	110.58 ± 26.43	111.25 ± 38.70	124.17 ± 37.51	118.90 ± 30.75		
LDL post BS	104.92 ± 28.19	117.47 ± 23.47	110.24 ± 29.11	109.37 ± 25.07		
LDL Final	90.42 ± 17.31 ^ab^	93.40 ± 17.20 ^ab^	111.95 ± 29.99	85.71 ± 22.86 ^ab^	0.003	0.153 ^#^
Cholesterol Admission	189.25 ± 33.27	195.07 ± 40.57	227.00 ± 38.49 ^a^	187.90 ± 32.13		
Cholesterol Release	178.92 ± 22.85	178.90 ± 31.53	197.60 ± 18.74	185.93 ± 34.19		
Cholesterol post BS	153.92 ± 21.99 ^ab^	162.27 ± 38.45	171.71 ± 31.94	180.60 ± 35.45		
Cholesterol final	155.50 ± 18.79 ^a^	156.47 ± 28.68	171.95 ± 35.04	146.50 ± 34.30 ^a^	0.001	0.234 ^#^
Triglycerides Admission	150.80 ± 33.94	164.40 ± 38.33	139.43 ± 20.09	152.35 ± 57.67		
triglycerides Release	130.42 ± 25.67	132.86 ± 30.65	118.85 ± 18.07	141.97 ± 68.16		
triglycerides post BS	92.33 ± 14.11	95.60 ± 31.45	99.52 ± 24.13	126.30 ± 41.03		
triglycerides Final	87.42 ± 24.44 ^ab^	80.47 ± 12.51 ^ab^	79.86 ± 16.82 ^ab^	85.50 ± 31.99 ^ab^	0.001	0.196 ^#^
Blood glucose Admission	103.75 ± 32.52	101.84 ± 30.60	100.43 ± 20.46	106.40 ± 37.73		
Blood glucose Release	111.17 ± 45.71	109.83 ± 27.31	89.50 ± 12.58	93.13 ± 17.62		
Blood glucose post BS	92.58 ± 16.62	83.20 ± 7.85	86.10 ± 10.74	90.83 ± 25.60		
Blood glucose Final	95.33 ± 21.27	82.73 ± 11.29	82.48 ± 8.48	79.57 ± 7.51 ^ab^	0.039	0.109 *

*p* < 0.05 (one-way ANOVA and post-hoc Bonferroni test). * Small effect, ^#^ Large effect. a: different compared with other timepoints within the groups (*p* < 0.001). b: Indicates a difference in relation to the other groups and time points (*p* < 0.001).BS: Bariatric surgery; BS2: Up to 2 years after bariatric surgery; BS4: Up to 4 years after bariatric surgery; BS6: Up to 6 years after bariatric surgery; BS+6: More than 6 years after bariatric surgery, Rel: Released after 3 months of treatment.

**Table 5 ijerph-16-01975-t005:** Frequency of comorbidities (diabetes, dyslipidemia, and hypertension) over time in the groups, according to the ACRO score (≤2.0).

**Diabetes Mellitus**	***n***	**At Admission**	**Release after BS**	**12 Weeks**	**24 Weeks**	**52 Weeks**
BS2	12	6	6	10	11	11
BS4	14	12	11	13	13	13
BS6	22	18	20	21	21	22
BS+6	30	21	22	30	30	30
**Dyslipidemia**	***n***	**At admission**	**Release after BS**	**12 weeks**	**24 weeks**	**52 weeks**
BS 2	12	6	7	10	11	12
BS 4	14	12	11	14	14	14
BS 6	22	14	16	20	20	20
BS +6	30	11	13	28	29	30
**Hypertension**	***n***	**At admission**	**Release after BS**	**12 weeks**	**24 weeks**	**52 weeks**
BS 2	12	1	1	7	7	8
BS 4	14	9	4	10	12	13
BS 6	22	9	11	17	18	18
BS +6	30	7	7	26	26	26

BS: Bariatric Surgery; The numbers represent the frequency and number of diagnosed subjects in each comorbidity over time [32].

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
