# Peer review of "Assessment of Cardiometabolic Risk Factors, Physical Activity Levels, and Quality of Life in Stratified Groups up to 10 Years after Bariatric Surgery"

_ijerph, 2019, doi:10.3390/ijerph16111975_

Round 1

Reviewer 1 Report

The manuscript by Costa et al aimed to investigate the levels of activity,  quality of life, clinical parameters, laboratory parameters and cardiometabolic risk factors after bariatric surgery (BS). The authors enrolled 78 patients submitted to BS, divided into four groups: evaluated 1–2 years  after BS (BS2), 2–4 years BS (BS4), 4–6 years after BS (BS6), and 6–10 years after BS (BS+6).They concluded that BS promoted improved physical activity, QOF and reduced comorbidities in patients with morbid obesity. The manuscript is very interesting although I've raised several issues:

1) The authors should decide if they want to report the number of each parameters in the text. If so, they should report for all variables or for none.

2) Instead of "in BS", I suggest to write "after BS" (line 26, page 1)

3) I suggest to analyze the data according to gender differences and to divide the population in men, women before and after menopause ( see and quote the following manuscript Vryonidou A et al EJE 2015)

4) I suggest to not use terms like "very good" or "excellent" ( see " The  quality of life of BS2 and BS4 was excellent, and for BS4 and BS+6 it was very good). They should reported numbers that demonstrate an improvement of that parameters

5) The improvement of cardiometabolic risk factors could be due also to the recovery of GH deficiency (see and quote Savastano S et al Growth Horm IGF Res. 2014) after bariatric surgery. This should  at least be reported in the discussion and provide also this possible explanation for the findings of the manuscript.

6) An extensive english editing  is required

Author Response

Reviewer 1

The manuscript by Costa et al aimed to investigate the levels of activity,  quality of life, clinical parameters, laboratory parameters and cardiometabolic risk factors after bariatric surgery (BS). The authors enrolled 78 patients submitted to BS, divided into four groups: evaluated 1–2 years  after BS (BS2), 2–4 years BS (BS4), 4–6 years after BS (BS6), and 6–10 years after BS (BS+6). They concluded that BS promoted improved physical activity, QOF and reduced comorbidities in patients with morbid obesity. The manuscript is very interesting although I've raised several issues:

1) The authors should decide if they want to report the number of each parameters in the text. If so, they should report for all variables or for none.

It has been described for a better understanding.

2) Instead of "in BS", I suggest to write "after BS" (line 26, page 1)

With the updated version of the English language, the abstract has changed. So this part has a better understanding.

3) I suggest to analyze the data according to gender differences and to divide the population in men, women before and after menopause (see and quote the following manuscript Vryonidou A et al EJE 2015)

2 paragraphs have been inserted to better exemplify what was requested.

4) I suggest to not use terms like "very good" or "excellent" ( see " The  quality of life of BS2 and BS4 was excellent, and for BS4 and BS+6 it was very good). They should reported numbers that demonstrate an improvement of that parameters

It has been modified as requested.

5) The improvement of cardiometabolic risk factors could be due also to the recovery of GH deficiency (see and quote Savastano S et al Growth Horm IGF Res. 2014) after bariatric surgery. This should  at least be reported in the discussion and provide also this possible explanation for the findings of the manuscript.

2 paragraphs have been inserted to better exemplify what was requested.

6) An extensive english editing  is required.

The whole article went through an extensive English editing for a better understanding.

Reviewer 2 Report

The manuscript “Assessment of cardiometabolic risk factors, physical activity level and quality of life in stratified groups up to 10 years after bariatric surgery” is an interesting report of a cross sectional follow up of bariatric surgery patients on factors related to their metabolic and psychological health. Overall, the manuscript is well-written and reports interesting findings, however, there are a few concerns.

1. In the method section, state that the study is a cross-sectional and not a longitudinal study. 

2. Clarify whether the age reported in Table 1 is age at surgery or age at study point (i.e. 10 years post surgery)

Provide more data on the follow up visit (i.e. post-surgery time point). It is evident that there are some retrospective measures and it is stated that questionnaires, etc. are given at the time point of interest. Data from biochemical measures are included for the post-surgery time point of interest, do the participants come to clinic and give blood? Were these values taken from medical records, etc.

3. Add percent weight loss to Table 3, instead of as Fig 2. 

4. For figures, group labels on the axis should match group labels, so not 2 years, but BS2.

5. Table 5. Overall, this data needs to be described more clearly. It is hard to follow what the numbers in the table are supposed to represent and therefore hard to evaluate the extent of the effects of BS on the frequency of comorbidities. 

Author Response

Reviewer 2

The manuscript “Assessment of cardiometabolic risk factors, physical activity level and quality of life in stratified groups up to 10 years after bariatric surgery” is an interesting report of a cross sectional follow up of bariatric surgery patients on factors related to their metabolic and psychological health. Overall, the manuscript is well-written and reports interesting findings, however, there are a few concerns.

1.         In the method section, state that the study is a cross-sectional and not a longitudinal study. 

The sentence was adjusted as requested.

2.         Clarify whether the age reported in Table 1 is age at surgery or age at study point (i.e. 10 years post surgery)

Provide more data on the follow up visit (i.e. post-surgery time point). It is evident that there are some retrospective measures and it is stated that questionnaires, etc. are given at the time point of interest. Data from biochemical measures are included for the post-surgery time point of interest, do the participants come to clinic and give blood? Were these values taken from medical records, etc.

A sentence has been inserted to better exemplify what was requested.

3.         Add percent weight loss to Table 3, instead of as Fig 2.

It has been modified as requested.

4.         For figures, group labels on the axis should match group labels, so not 2 years, but BS2.

It has been modified as requested.

5.         Table 5. Overall, this data needs to be described more clearly. It is hard to follow what the numbers in the table are supposed to represent and therefore hard to evaluate the extent of the effects of BS on the frequency of comorbidities. 

It has been modified as requested.

Round 2

Reviewer 1 Report

The authors have addressed all comments adequately. As a result, I now recommend the current form of the manuscript can be accepted for publication without further modification.